# Processing Speed and Time since Diagnosis Predict Adaptive Functioning Measured with WeeFIM in Pediatric Brain Tumor Survivors

**DOI:** 10.3390/cancers13194776

**Published:** 2021-09-24

**Authors:** Maria Chiara Oprandi, Viola Oldrati, Morena delle Fave, Daniele Panzeri, Lorenza Gandola, Maura Massimino, Alessandra Bardoni, Geraldina Poggi

**Affiliations:** 1Neuro-Oncological and Neuropsychological Rehabilitation Unit, Scientific Institute, IRCCS E. Medea, Bosisio Parini, 23842 Lecco, Italy; viola.oldrati@lanostrafamiglia.it (V.O.); morena.dellefave@lanostrafamiglia.it (M.d.F.); daniele.panzeri@lanostrafamiglia.it (D.P.); alessandra.bardoni@lanostrafamiglia.it (A.B.); geraldina.poggi@lanostrafamiglia.it (G.P.); 2Department of Medical Oncology and Hematology, Pediatrics Unit, Fondazione IRCCS Istituto Nazionale dei Tumori, 20133 Milan, Italy; lorenza.gandola@istitutotumori.mi.it (L.G.); Maura.Massimino@istitutotumori.mi.it (M.M.)

**Keywords:** pediatric brain tumor, adaptive functioning, processing speed, hydrocephalus, rehabilitation, time since diagnosis, WeeFIM

## Abstract

**Simple Summary:**

Brain tumor (BT) survivors show difficulties in adaptive functioning (AF) and in acquiring independence (e.g., graduating, finding employment, building strong relationships, and being independent). The aim of our observational retrospective study is to explore the contribution of different clinical and cognitive variables in explaining and predicting the AF outcomes of BT survivors, measured with the Functional Independence Measure for Children (WeeFIM). The analysis demonstrated that processing speed and time since diagnosis are the main explanatory variables. Other clinical factors, such as age at diagnosis and hydrocephalus, differentially influence functional skills according to distinct domains (i.e., self-care, mobility, and cognition). The identification of the clinical factors influencing AF could suggest targets on which to focus attention. By successfully assessing, understanding, and managing AF, it will be possible to improve its management in pediatric BT survivors.

**Abstract:**

(1) Background: Brain tumor (BT) survivors show difficulties in the acquisition of developmental milestones, related to academic achievement, vocational employment, social relationships, and autonomy. The skills underlying adaptive functioning (AF) are usually damaged in BT survivors due to the presence of the brain tumor, treatment-related factors, and other neurological sequelae. In this study, we aimed to explore the contribution of different cognitive factors in children with BT to AF, considering diagnosis-related variables. (2) Methods: Standardized cognitive assessment was undertaken and clinical information was collected from a retrospective cohort of 78 children with a BT, aged between 6 and 18 year old at the time of the assessment. Regression models were computed to investigate the influence of the selected variables on daily functional skills as measured by the Functional Independence Measure for Children (WeeFIM). (3) Results: The analyses showed that the main explanatory variables are processing speed and time since diagnosis. Other clinical variables, such as age at diagnosis and hydrocephalus, differentially influence functional skills according to distinct domains (i.e., self-care, mobility, and cognition). (4) Conclusions: The main explanatory variables of AF that emerged in our models point to a potential target of improving AF management in pediatric BT survivors.

## 1. Introduction

Adaptive functioning (AF) is defined as a person’s ability to manage the demands of everyday life [1]. The complexity of these challenges increases progressively in children [2], while several cognitive and behavioral abilities improve with age [3]. The American Association on Intellectual and Developmental Disabilities (AAIDD) divides AF into three different sub-domains: conceptual abilities (e.g., language, communication, reading, and writing), practical skills (e.g., self-care, nutrition, and dressing), and social competencies (e.g., interacting with other people and acting appropriately in a social context) [1,4]. 

The skills underlying AF are a combination of cognitive abilities such as attention, processing speed, and working memory [5]. These domains are usually damaged in brain tumor (BT) survivors due to the presence of the tumor; treatment-related factors; and other neurological sequelae, particularly hydrocephalus [6,7,8]. 

These complicating factors lead BT survivors to fail to reach the main developmental milestones, such as finishing their educational career (i.e., graduating from high school and university), achieving vocational satisfaction, having lasting relationships, maintaining autonomy, and coping with daily living activities; overall, they have difficulties being independent [9]. BT survivors have a significantly lower rate of independence in adulthood than other cancer survivors (e.g., Hodgkin’s lymphoma) [10].

A growing body of evidence suggests that cognitive skills may act as mediators between clinical variables and AF in pediatric patients with acquired brain injury (ABI) [7,11,12,13], but research on the cognitive predictors of AF in pediatric BT patients is scarce. 

Papazoglou et al. demonstrated that attention span predicted the communication score in a cerebellar population of pediatric BT [14]. Semmel et al. found a significant indirect effect in a BT sample between the cognitive predictors of attention span and processing speed (PS), and neurological condition and AF; taken together, however, the neurological risk and PS had the power to explain 39% of the variance in the AF scores [15].

PS is the speed at which mental operations are performed [16], and it is one of the most compromised functions in pediatric BT patients [7], even when cognitive functioning falls within the average range [17]. PS deterioration is mainly caused by a disruption in the white matter, which is extremely important for the diffusion of action potentials, but it is also particularly vulnerable to toxic and physical agents, such as radiation [18,19]. Research on pediatric traumatic brain injury (TBI) patients suggests that PS underlies AF abilities [13,15] and that PS is an important mediator between AF and TBI [12]. 

Among the clinical factors associated with the poor AF outcomes found in pediatric BT patients, time since diagnosis (the period between the age at diagnosis and the age at assessment) [9,20,21], age at diagnosis [21,22], histopathological type of tumor [23,24,25], hydrocephalus [4,26], and tumor location [23,26] have been mentioned.

Other clinical factors associated with poor AF in BT survivors are related to oncological treatments, namely, exposure to radiation therapy, particularly cranial irradiation; recurrence [27,28]; subtotal resection; chemotherapy [26,29,30,31]; and additional surgery [32].

Studies on AF predictors in BT survivors have used different tools to assess these abilities, such as the Scales of Independent Behavior-Revised [5,25], the Vineland Adaptive Behavior Scale [14], and the Adaptive Behavior Assessment System—Second Edition [4]. These tools are based on interviews administered by an examiner to a parent or a caregiver [33]. Although the measures obtained with these interviews are generally reliable and stable, they may be subject to bias and influenced by several factors related not only to the child but also to the parents [26].

One of the gold standard tools used to measure the AF of young patients with disabilities is the Functional Independence Measure for Children (WeeFIM). This tool was directly adapted from the Functional Independence Measure (FIM) for adult patients [34] developed from the National Task Force for Medical Rehabilitation in 1983 [35]. WeeFIM was built on the disability conceptual model of the World Health Organization, considering the concepts of “pathology, impairment, disability, and handicap” and of the “burden of care” [36,37]. WeeFIM was selected by the Common Data Elements Traumatic Brain Injury Outcomes Workgroup as a tool recommended for measuring AF and daily living competencies; it has strong statistical properties in terms of reliability and validity (see Section 2.2.2 for details) and a strong capacity to provide information about functional status and to detect changes after rehabilitation intervention [38].

The predictors of WeeFIM outcomes have been explored in children with TBI [39,40,41,42,43] or in other clinical populations, such as patients with active epilepsy [44] or those diagnosed with anti-N-methyl-D-aspartate receptor encephalitis [45] or myelomeningocele pediatric patients [46].

In contrast, to the best of our knowledge, no studies have been conducted on the predictors of WeeFIM outcomes in BT survivors, only in ABI population (with samples not including BT) [43,47,48]. These studies reported significant correlations between neuropsychological standardized tests (assessing memory, language, and cognitive functioning) and the WeeFIM total score and/or its subscales [43,47,48].

Due to several risk factors being simultaneously present in BT survivors, it is difficult to evaluate the effect of each single variable [9]. Despite the importance of exploring AF predictors to identify higher-risk patients as early as possible and to properly set rehabilitation programs, this topic is under-investigated in BT pediatric patients. Therefore, the current observational retrospective study was aimed at investigating the predictors of WeeFIM outcomes in a sample of pediatric BT survivors to better understand how they affect AF in this population.

## 2. Materials and Methods

### 2.1. Participants

A total of 73 patients (*n* = 42 male patients, 57.5%) were obtained after the removal of outliers (see Section 2.2.5 for details). This retrospective cohort included patients between 6 and 18 years of age at assessment with a diagnosis of BT treated at a pediatric rehabilitation center in Italy (Scientific Institute I.R.C.C.S. E. Medea, Bosisio Parini, Italy). The clinical characteristics of the patients are listed in Table 1.

Participants were eligible for the research if they were (i) diagnosed with a BT (in oncological institutions other than our center, after undergoing an MRI, a biopsy procedure, and with the concomitant opinion of an oncologist); (ii) aged between 6 and 18 years at assessment; (iii) mentally and physically able to undergo cognitive assessment; and (iv) assessed with all of the selected cognitive and AF measures. The exclusion criteria were the presence of a pre-existing neurodevelopmental disorder or disability, the diagnosis of a pervasive developmental disorder, or the diagnosis of neurofibromatosis [49].

Patients underwent both cognitive and AF assessment during the same hospitalization, the former with a neuropsychologist and the latter with a physiotherapist who had completed the appropriate WeeFIM certification.

Approval was received from the local ethical standards committee on human experimentation at the Scientific Institute I.R.C.C.S. E. Medea. The study was conducted in agreement with the principles expressed in the 1964 Declaration of Helsinki. Due to the observational nature of the study, the Ethics Committee of Scientific Institute I.R.C.C.S. E. Medea only required notification about the study (identification number: 03.2021 Oss; date of approval: 14 April 2021), and written informed consent was not required from the parents/caregivers prior to study enrollment.

### 2.2. Measures

#### 2.2.1. Demographic and Clinical Information

Information on the medical variables were collected from the patients’ charts. Alongside age at diagnosis, the time since diagnosis was collected, reflecting the time (expressed in months) from the diagnosis to the functional evaluation.

The histopathological type of tumor was classified as astrocytoma (all pilocytic in our sample), ependymoma, medulloblastoma, or mixed diagnosis (such as primitive neuro-ectodermal tumors (PNET), craniopharingioma, glioblastoma, atypical teratoid rhabdoid tumor, choroid plexus tumor, pineoblastoma, or brainstem glioma). History of hydrocephalus was recorded as present or not present. Tumor location was rated as supra- or infratentorial. Treatment type was classified as neurosurgery alone without adjuvant treatments, neurosurgery and chemotherapy, and neurosurgery and radiotherapy without or without chemotherapy. Two patients who underwent both adjuvant treatments but did not undergo neurosurgery were moved into the neurosurgery and radiotherapy with or without chemotherapy treatment group. This choice was made as it allows us to examine the impact of radiation as an independent clinical variable.

#### 2.2.2. AF Assessment

The WeeFIM (version 5.0, University of Buffalo, Buffalo, NY, USA) [49] is an international scale that assesses pediatric disability, measuring and quantifying the level of assistance or supervision required for daily tasks [50]. The tool is used in patients with disability from 6 months to 21 years of age [35]. It is composed of a list of 18 activities of daily living, in which scoring returns a total score and three sub-scores: self-care, mobility, and cognition. These three subdomains do not fully correspond to the domains as specified by the AAIDD (see Section 1). Nevertheless, the items of the WeeFIM self-care and mobility scales address competencies similar to those covered by the AAIDD practical skills domain. Similarly, the WeeFIM cognition items match the AAIDD’s conceptual abilities, except for the social interaction item, which could fall into the AAIDD’s social competencies domain.

Each item is rated on a seven-point scale on three levels of dependence/independence:Complete dependence: 1 = total assistance (subject = 0%–24%); 2 = maximal assistance (subject = 25%–49%);Modified dependence: 3 = moderate assistance (subject = 50% or more); 4 = minimal contact assistance (subject = 75% or more); 5 = supervision;Independence: 6 = modified independence (with device(s)); 7 = complete independence (no device, completing the task promptly and safely) [37].

The lowest possible total score is 18 and the highest is 126, with a lower score indicating less autonomy and more need for assistance [39]. WeeFIM is relatively short to administer: it only takes 15–20 min to interview a caregiver with a trained administrator [38].

In accordance with Suskauer et al., a good outcome on the overall scale is scored with 85 or more; between 75 and 84, the score is considered moderate; and a poor performance is a score that falls below 70 [42,51].

WeeFIM Developmental Functional Quotients were used to provide a standard score of age-appropriate functioning to allow for comparison across age groups [34].

WeeFIM has been widely studied, and the evidence is strong regarding its reliability and validity: the internal consistency Cronbach’s α is 0.90, the interrater interclass correlation falls between 0.73 and 0.94, and test–retest interclass correlation is 0.97 [38].

#### 2.2.3. Cognitive Assessment

Patients underwent an assessment of cognitive functioning with the administration of the Wechsler Intelligence Scale for Children, 4th Edition (WISC-IV). Performance yields five summary scores:The Verbal Comprehension Index (VCI) assesses verbal reasoning skills;The Perceptual Reasoning Index (PRI) measures visual-spatial reasoning skills;The Full Scale Intelligence Quotient (FSIQ) is the sum of the two previous indices and a measure of overall intellectual functioning;The Processing Speed Index (PSI) is a measure of the ability to respond promptly and to focus attention on a task; andThe Working Memory Index (WMI) is a measure of auditory attention, concentration, and mental manipulation of information in short-term memory.

Scores are reported as age-corrected standard scores with a mean of 100 and a standard deviation of 15. Lower scores represent worse performance [52].

#### 2.2.4. Selection of the Explanatory Variables

Regarding the cognitive variables, we decided to examine the explanatory effect of PS only, as preliminary analysis showed strong correlations (all *r* ≥ 0.50, *p* < 0.001) between all the WISC-IV indexes and PS, which may have led to multicollinearity problems. Our decision was also based on the relevant literature about the key role of PS in BT survivors and its deterioration, even when cognitive functioning is preserved [16,17,53,54].

Regarding the clinical characteristics, time since diagnosis, age at diagnosis, histopathological type of tumor, history of hydrocephalus, tumor location, and treatments were selected as the explanatory variables given previous evidence (see Section 1).

Finally, the subscales of the WeeFIM were selected as dependent variables, with each one included in a separate regression model, as they reflect specific AF domains. The total scale consisting in the sum of the subscale scores was used as a generic proxy of AF but lacks the specificity reflected by the subscales and was thus not selected as a dependent variable of interest. 

#### 2.2.5. Data Diagnostics and Statistical Analysis

First, three separate general linear regression models were computed on an initial sample of 78 participants to estimate each WeeFIM subscale (i.e., self-care, mobility, and cognition) from a set of seven explanatory variables: (1) PS index from WISC-IV, (2) time since diagnosis (in months), (3) age at diagnosis (in months), (4) histopathological type of tumor (astrocytoma vs. ependymoma vs. medulloblastoma vs. mixed diagnosis), (5) history of hydrocephalus (present vs. absent), (6) tumor location (supratentorial vs. infratentorial), and (7) treatments (neurosurgery alone, neurosurgery and chemotherapy, and neurosurgery and radiation with or without chemotherapy). A sample of 78 participants was considered sufficient to estimate models with seven explanatory variables according to the rule of thumb, suggesting 10/15 observations per number of predictors [55].

Diagnostic tests were conducted on the three models to check that regression assumptions were met. Visual inspection of quantile–quantile (Q–Q) plots created for each WeeFIM subscale indicated the presence of five potential outlier observations. These outliers were then identified as participants’ scores falling below 3 SD from the mean of the dependent variables, corresponding to two participants under-performing in both the self-care and mobility subscales and the other three participants under-performing each on a different subscale. The removal of outliers left a final sample of 73 participants. Further diagnostic tests of the three models computed on the final sample confirmed the assumptions of the independence of errors (using the Durbin–Watson test, all *p* > 0.6) and heteroscedasticity (using the gvlma package in R) [56] being met. However, both distributions of the mobility and the cognition subscales were found to be negatively skewed [50,52]. Additionally, CERES plots suggested a possible violation of linearity in all three models. Given the results of the diagnostic tests, we decided to compute generalized additive models (GAMs), as they allow the relationships between the explanatory variables and the dependent variable to be nonlinear and to be described by smooth curves [57]. The restricted maximum likelihood (REML) smoothing parameter estimation was set to prevent both excess wiggliness (overfit of the model) and excess smoothness (underfit of the model) [58]. Thus, GAMs with smoothed splines functions applied to diagnosis time and PS variables, including the three categorical variables (tumor location, type, and history of hydrocephalus), were computed for each WeeFIM subscale. These functions are associated with estimated degrees of freedom (EDFs), which indicate whether the relationship between the variable is either linear (EDF = 1) or nonlinear (EDF > 1). Furthermore, diagnostic tests were performed on the GAMs to assess the goodness of fit to the data.

Finally, we performed a mediation analysis to test whether the PS index may mediate the effects exerted by the clinical factors included in the models (i.e., time since diagnosis, tumor location, tumor type, history of hydrocephalus, and treatments) on the three subscale scores. The significance of the estimated causal mediation effect of the PS index was tested using nonparametric bootstrapping procedures. Unstandardized indirect effects were computed for each of the 10,000 bootstrapped samples. Age at diagnosis was included as a predictor variable alongside the other clinical factors in the mediation models computed for each WeeFIM subscale.

The level of statistical significance in all tests was defined as *p* < 0.05. R software (version 4.0.3; R Foundation for Statistical Computing) was used to perform all of the statistical analyses in this article, and the mgcv package was used for GAM estimation [59]. Mediation analysis was performed with the mediation package [60].

## 3. Results

The means and standard deviations of the WISC-IV indexes of our sample are reported in Table 2. All of the measures fell within the average range, except for the PS index, which fell in the borderline range.

Three separate GAMs were estimated for the self-care, mobility, and cognition subscales. Table 3 provides a brief summary of the models’ characteristics.

Regarding the self-care model, no parametric term was found to be statistically significant (all *p* > 0.05). Regarding the smooth terms, the self-care model showed that the effect of diagnosis time was significant but nonlinear (EDF = 2.6, F = 10.46, *p* < 0.0001), indicating better self-care skills with increasing time from diagnosis to functional evaluation, with a more accentuated increase within the first 100 months from diagnosis. The effect of PS was also found to be significant and nonlinear (EDF = 2.4, F = 3.47, *p* < 0.0001). As illustrated in Figure 1, self-care skills increase with a higher PS especially in the inferior-range of the PS (i.e., score < 80); when approaching the normal range of the PS, the increase becomes less accentuated. The analysis also showed that the effect of age at diagnosis was significant but roughly linear (EDF = 1.1, F = 4.3, *p* < 0.0001), indicating worse self-care skills in subjects with earlier diagnosis.

The mobility model showed the variable history of hydrocephalus to be statistically significant (coefficient = –2.65, SE = 1.04, *t* = –2.53, *p* < 0.02). Participants with a history of hydrocephalus (M = 30.00, SE = 0.41) displayed poorer mobility than participants without hydrocephalus (M = 32.6, SE = 1.63). No other parametric term was significant (all other *p* > 0.05). With regard to the smooth terms, the mobility model showed that the effect of both diagnosis time (EDF = 1.1, F = 1.96, *p* < 0.0001) and the PS (EDF = 0.82, F = 0.52, *p* < 0.02) were significant and linear, indicating better mobility skills with increasing time from diagnosis to functional evaluation and a higher PS (Figure 1). Conversely, the association between mobility and age at diagnosis was not significant (*p* = 0.5).

In the cognition model, no parametric term was found to be statistically significant (all *p* > 0.05). Examining the smooth terms, the cognition model showed that the effect of diagnosis time was significant and linear (EDF = 0.87, F = 0.77, *p* < 0.01), thus suggesting better cognitive skills with increasing time from diagnosis to functional evaluation. The effect of the PS was also significant and roughly linear (EDF = 1.3, F = 1.7, *p* < 0.001), indicating better cognitive skills with a higher PS (Figure 1). The effect of age at diagnosis was not significant (*p* = 0.8).

Lastly, the diagnostic functions provided by the mgcv package in R software confirmed that full convergence was obtained for all models, ruling out the inclusion of too many parameters in each model and that residuals were randomly distributed for both the smooth terms in all models. Figure 2 depicts the sample’s mean scores for each subscale for the WeeFIM items.

Finally, the results of the mediation analyses revealed that the PS did not mediate the effects of the clinical factors, namely, time since diagnosis, tumor location, type of tumor, history of hydrocephalus, and treatments, on any of the WeeFIM subscales (all *p* > 0.2).

## 4. Discussion

The purpose of this study was to examine the effect of the relevant cognitive and clinical variables on AF outcomes in pediatric BT survivors.

The results demonstrated that PS and time since diagnosis are the factors that may explain BT survivor outcomes in all three specific AF domains, measured with the WeeFIM. Other variables were found to differentially influence specific AF domains: age at diagnosis for the self-care subscale and history of hydrocephalus for the mobility subscale. Ultimately, mediation analysis showed that PS did not mediate the other clinical factors. The lack of a significant mediational effect suggests that the significance of time since diagnosis and history of hydrocephalus emerged according to the specific AF domains is not influenced by the level of PS.

The significant explanatory variables that arose in our models are the focus of this discussion.

### 4.1. PS Effects on AF

PS proved to be a significant explanatory factor in our models of AF outcomes. The relationship between PS and AF has also been reported in other clinical populations, such as in TBI [13], autism patients [61,62], and children with attention deficit and hyperactivity disorder [63,64].

The relationship between PS and the cognition subscale is not surprising: PS is a core function that underlies other higher-level cognitive skills [17]. A neurodevelopmental model of BT children’s long-term outcomes supposed that a deficit mainly driven by a slow PS, with a cascade effect causing working memory and attention difficulties, is the main cause of poor cognitive outcomes [7,65]. PS was found to be vulnerable to treatments directly on the central nervous system, such as cranial radiotherapy, which cause reduced, normal-appearing white matter volume and leads to cognitive dysfunctions [17].

PS is also a good predictor of the self-care subscale. Thornton et al. found similar results in cancer survivors: Reductions in PS were related to negative outcomes in practical skills, such as managing the home, taking care of personal hygiene, and planning [53]. It is likely that delayed PS creates difficulties in understanding verbal instructions and in keeping up in settings that are fast-paced and highly demanding [12]. Importantly, self-care skills have been found to be one of the best predictors of independent living, post-secondary education (i.e., attending college or university) [66], and employment in people with disabilities [66,67].

The items composing the self-care subscale have a strong motor component (i.e., this domain includes activities such as washing, eating, and dressing), and PS was demonstrated to also play an important role in visual-motor performance, with worse behavioral responses associated with delayed PS [68].

PS is also a predictor of the mobility subscale. It has been proposed that a disruption in white matter tract integrity might interrupt communication between the areas involved in the appropriate planning and monitoring of movement, causing motor impairment [69].

Aukema et al. studied the relationship between the disruption of white matter, PS, and motor speed in a pediatric medulloblastoma group [70]. They found that white matter in the right inferior fronto-occipital fasciculus, the splenium, and body of the corpus callosum showed a positive correlation with PS and motor speed [70].

Notably, our analysis demonstrated that the relationship between PS and the mobility and cognition subscale scores is linear: PS and WeeFIM outcomes in these two domains constantly increase together (Figure 1). However, the relationship between PS and the self-care subscale is almost quadratic: The self-care outcome increases more steeply until the PS score was around 80–84 (Figure 1), from which point onward its increase becomes less accentuated. This finding seems to suggest that PS evaluation is more informative regarding patient independence when the score is below 80–84, which in the WISC-IV represents the lower limit for intelligence, indicating borderline intellectual functioning [52].

### 4.2. Clinical Variable Effects on AF

#### 4.2.1. Time since Diagnosis

The controversial literature about time since diagnosis suggests that the comprehension of its contribution should be studied in more depth.

Our results showed that the AF scores of BT survivors improved with increasing time since diagnosis, in accordance with a few studies reporting similar enhancements in domains such as cognitive functioning [71], motor speed, and dexterity [72].

In our opinion, this improvement might be explained by several factors.

First, the longer the time since diagnosis, the weaker the acute effects of tumor treatments and their implications [73]. The active treatment period usually lasts approximately a year [74], with a long hospitalization, a possible prolonged stay away from home, frequent irregular attendance at school, and consequent separation from peers [75].

In accordance with some studies examining AF in the acute phase of treatment in BT survivors, self-care functioning was found to be one of the most impaired areas a few weeks after the surgery, with difficulties persisting for up to six months [75]. This is understandable because active treatments more heavily impact everyday skills [23].

Spiegler et al. found similar results in cognitive functioning, suggesting that, after an initial decline in the cognitive processes during the early period after the end of treatment, a slowdown in the decrease occurs [72].

Second, the longer the time since diagnosis, the more rehabilitation interventions the patient has undergone. Based on the severity, the phase of the disease, and its consequences, rehabilitation interventions can be offered with different modalities and in accordance with the patient’s needs. Usually, an inpatient pediatric rehabilitation setting includes a multi-professional approach, with physical, occupational, and speech therapists; neuropsychologists; and psychotherapists, for at least three hours per day [76] for several weeks [77]. The hypothesis that longer time since diagnosis corresponds to a higher number of rehabilitation interventions that BT survivors have undergone is purely speculative, and further investigations are needed to explore this potential relationship.

Third, the longer the time since diagnosis, the greater the parental acceptance of their child’s disease and its consequences. Notably, WeeFIM scores are subjective parental evaluations of their children’s AF. Acceptance may be crucial for the caregiver’s adjustment; it allows for a better psychological flexibility, which is necessary in pediatric oncology [78,79], where the clinical situation of the patients during the active treatment phase is constantly changing, causing high stress [74].

A key role in perceived parental distress (or well-being) is played by their coping strategies. Evidence suggests that the responses of parents with children diagnosed with a BT evolve over time [74]. At the time of diagnosis, parents show a high use of guidance seeking and coping skills, demonstrating a need to be supported by other figures (such as professionals, friends, family, and spiritual advisors). Over time, the patients’ mothers showed an increase in problem solving skills and in both parents’ acceptance and resignation, with beliefs such as accepting that things will not return to how they used to be before the diagnosis, that time would not make a difference, and that the situation cannot be controlled. These trajectories suggest the use of more reasonable coping skills among primary caregivers over time, even if emotional distress remains understandably high [74]. These findings seem to fall in line with evidence of decreasing levels of distress in both children with cancer and their parents over time [80,81].

Although intellectual functioning and some neuropsychological skills often show a decline in BT survivors, IQ and AF may not follow the same trajectories of development [26]. Some studies have suggested that, in healthy children and those with intellectual impairment, AF and IQ are associated but with small or moderate effects. Therefore, these constructs are related but not overlapping, and in ABI patients, including those with a BT, this relationship is also less predictable [26,82]. Netson et al. found that at five years post-diagnosis, the percentage of patients with an AF score below the average tended to diminish, while the trajectories for the IQ score kept falling below the average [26]. A further factor to consider is that, after the end of active treatment, when patients are discharged from the hospital, they find better structured environments, both at school and at home. The routine can help them learn the behaviors necessary to manage everyday demands, with continuous repetition. It is possible that this stability makes them more independent in their environment, although their difficulties may be highlighted in less predictable contexts.

Notably, mean time since diagnosis in our sample was 59.5 months (4.9 years). Five years is usually cited in many articles as a crucial time in the survival rate. Today, the survival for pediatric BT is around 70–80% five years post-diagnosis in most European countries [83]. The 20–30% who do not survive for five years is probably represented by children with more aggressive tumors who suffer from more severe consequences, which may, in turn, worsen AF outcomes.

In closing, similar to that which accounted for PS, time since diagnosis also showed a nonlinear relationship with the self-care subscale: The outcome increased more steeply until the time since diagnosis was around 100 months (almost eight years; Figure 1), from which point onward, its increase appeared to slow. This finding seems to suggest that a temporal window of several years after the diagnosis is present, in which different dynamics, such as those already mentioned, may intervene and influence the self-care outcome of BT survivors. 

#### 4.2.2. Age at Diagnosis

We found age at diagnosis to be a significant predictor in the self-care model. A younger age at diagnosis is usually associated with worse cognitive functioning [84] and poorer AF [21]. This pattern was present in our sample as well. The young brain and white matter maturation are particularly sensitive to the effects of toxic agents, such as radiotherapy [19,85]. Preserved PS functioning is mainly driven by the health status of the white matter, and as mentioned before, reductions in PS are related to negative outcomes in practical skills, including self-care [53]. Accordingly, treatment protocols tend to avoid or delay irradiation for children that are younger at diagnosis [85].

Our results are in line with those reported by Kunin-Batson et al. (2011); they found that young patients diagnosed with a brain tumor before the age of six were half as likely to be independent than those diagnosed at an older age (i.e., 12 years) [10].

Notably, in this subscale, both age at diagnosis and time since diagnosis were significant. The former is mechanistically related to the sensitivity of the developmental brain during the early stages of growth [86], whereas the latter may underlie the influence of adjustment processes (clinical and/or psychological). Contrary to findings for the self-care skills, age at diagnosis was not associated with mobility or cognition skills, suggesting that, although correlated, the two variables exert differential effects on these AF domains. This may be related to the self-care subscale being more investigated; with eight items, it may be better able to capture changes related to age at diagnosis. The other two subscales are composed of just five items and the evaluation may be less accurate.

#### 4.2.3. History of Hydrocephalus

Hydrocephalus was found to be one of the predictors on the WeeFIM mobility subscale. Today, its role as a significant risk factor for poor performance is well recognized [87]. Children with hydrocephalus perform worse in intelligence tests than healthy children and children with the same pathology but without hydrocephalus [88]. More precisely, children with hydrocephalus often show a high discrepancy between verbal IQ and their performance score, to the advantage of the former. Hydrocephalus severity is thought to negatively affect visuospatial [89] and fine-motor skills, such as manual speed and visuomotor coordination [90]. This is probably related to the diffuse white matter damage caused by the increased pressure in the brain [88,91].

The literature suggests that a good physical and motor functioning is an important factor for cancer survivors to find employment, to financially support themselves, and to therefore facilitate independent living [92].

### 4.3. BT Survivors’ AF Characteristics

An examination of the means of the eight individual items composing the self-care subscale showed that the majority of items fall in the range between 4 and 7 scores (Figure 2). Therefore, the BT survivors in our sample demonstrated the need for some level of assistance, ranging from a minimal contact assistance (score of 4) to supervision or prompts to set up a task (score of 5) or to the need for an assistive device (score of 6) in the majority of the self-care activities such as grooming, bathing, dressing, and toileting.

The variability of the observations of the mobility subscale was lower. All of the ratings fell in the range of 5–7, showing that the most difficult motor activities in this sample were tub/shower transfer and climbing the stairs, especially for those patients who developed hydrocephalus.

Higher ratings were registered in the cognition subscale, with all of the items approaching 7, except for the problem-solving item. Even though the cognition subscale and its single items were demonstrated to correlate with standard neuropsychological tests in a pediatric clinical population [47], the assessment of this domain by the WeeFIM is much less detailed, and it is likely that the evaluation is not refined enough to identify existing difficulties.

### 4.4. Parent Report Implications

Notably, contrary to cognitive functioning, AF assessment usually relies on parents’ reports. The most used AF scales, i.e., the Vineland Adaptive Behavior Scales, the Adaptive Behavior Assessment System-Second Edition, the Woodcock–Johnson Scales of Independent Behavior-Revised, the Diagnostic Adaptive Behavior Scale, and the WeeFIM, are all interviews administrated to caregivers, since AF skills need to be observed during the patient’s daily routine [93]. As a consequence, they are neither assessable directly by an outside examiner nor exclusively in clinical, non-ecological settings. Caregivers are privileged observers of AF, but they are subject to bias and may be influenced in their estimates by other factors, such as their coping skills and expectations.

Previous research has highlighted two main coping approaches towards oncological pediatric disease: problem- or emotion-focused [94,95]. The former is more active and involves overcoming problems with practical solutions to produce change; the latter is more passive and aims at avoiding stress, but without a real change in the situation, increasing the risk of developing anxiety and depressive symptoms [94,95]. All coping responses have limitations depending on the degree of flexibility with which they are applied. Problem-focused parents may be more active and sometimes more intrusive, and they tend to perform the activities the children should do themselves (i.e., wash or dressing themselves). Conversely, caregivers with an emotion-oriented approach may be more passive, even when the child may benefit from increased support. This difference in the approaches can certainly affect a parent’s evaluation of their child’s need for assistance, overestimating or underestimating their AF.

Other factors influencing parent AF evaluations are the expectations of their children’s performances, which previous research has shown to be one of the most significant predictors for the outcomes of children with disabilities [67,96]. The impact exerted by the parents’ involvement in their children’s outcomes may reflect their efforts in making goals more achievable for their children (i.e., helping and supporting them or asking for assistance) and/or it may reflect a biased evaluation of the children’s outcomes (i.e., an over- or underestimation of the child’s competencies).

### 4.5. Hypotheses of Intervention

The identification of the clinical factors influencing AF may indicate potential targets of intervention. Indeed, some of the factors identified are not modifiable, such as the age at diagnosis, but in accordance with the previous literature on BT survivors, it is crucial to take them into account to identify those patients at greatest risk of poor adjustment outcomes.

Other factors are partly modifiable, such as the presence of hydrocephalus or the processes of psychosocial adjustment occurring over time. For example, psychological intervention could be useful to improve parents’ and patients’ coping skills and their acceptance of the disease and its consequences [74]. The processing speed as well as related neuropsychological variables (i.e., working memory, attention and executive functions), could also represent a potential target for intervention. Cognitive rehabilitation has a twofold purpose: on one hand, it aims to restore impaired skills; hence, the implementation of specific training and repetition and practice of the skills involved, and on the other, it aims to improve the compensation for impaired functions [97]. There is evidence of improved PS after a training program for healthy children and adolescents [98]. However, cognitive rehabilitation intervention studies for pediatric BT children are scarce, even though the emerging results are encouraging [99]. Butler et al. [100,101] found positive effects on attention following an in-person, therapist-guided cognitive remediation program. Computerized home-based interventions have also proved to be effective in enhancing executive functions, including PS [102,103]. Unfortunately, whether these improvements can be generalized beyond the testing task phase to daily life functioning and quality of life is an issue that needs further investigation [99].

Compensatory measures should also be considered for pediatric BT patients in the school setting. Indeed, it has been suggested that adjunctive schooling efforts, coupled with cognitive rehabilitation interventions, may exert a prophylactic effect [101]. Goreman et al. suggested that TBI patients who show similar difficulties to BT children should be given extra time to complete both class tests and home assignments, possibly with a reduced length but the same contents. Moreover, they would benefit from smaller units containing less information to ease learning and from the use of assistive technologies to compensate for slowed PS [104].

### 4.6. Limitations

Our study is not devoid of limitations. The research was exploratory in nature; thus, replication is necessary to confirm our findings or to better investigate the role of some variables, such as treatment-related factors, with more precise information (e.g., radiation doses, type of radiotherapy that the patients underwent, and number of chemotherapy cycles) or other neuropsychological variables (e.g., attention, executive functions, and language). A larger sample would have allowed us to explore the interaction between variables.

Recruitment bias may be present in this research, as reported in similar studies [25]. Our sample only included the patients for which full cognitive and AF assessment was possible and available. However, subjects with poorer functioning (such as those that develop the posterior fossa syndrome) are often not able to complete the standard evaluation protocols. For this reason, they may have been automatically excluded from our sample.

Additionally, we relied on parents’ ratings of survivors’ functioning; thus, children’s functional performance should be more objectively evaluated to provide a less-biased estimation of their competencies. Further studies are needed to better investigate whether a real improvement in the AF performance of BT patients occurs or if the improvement is related to the caregiver’s acceptance of the disease and its consequences. An evaluation of parents’ expectations, parenting styles, and approach to diagnosis-related difficulties may help to disentangle perceived and effective AF difficulties experienced by BT survivors.

We conjecture that time since diagnosis predicted better AF outcomes because the longer the time since diagnosis, the more rehabilitation interventions the patient has possibly received. No information about this topic was further investigated. Therefore, this hypothesis is speculative and should be considered with caution.

We also note social competencies, which are listed among the most compromised AF skills in BT survivors with time elapsed since diagnosis [21,26]. Unfortunately, the WeeFIM scale only has one item related to this domain in the cognition subscale (social interaction); therefore, the assessment of these skills in our sample was not sufficiently investigated to draw any conclusion.

One additional limitation is the presence of an evident ceiling effect, a well-known shortcoming of the WeeFIM [38], which was also present in our sample, especially in the mobility and cognition subscales: 36 (49.31%) patients scored 48–56 (meaning a score of 6 or 7 in all of the items) in the self-care subscale, while 56 (76.71%) scored 30–35 in the motor and in the cognition subscales. This could explain the different variability of the three domains (See Figure 1), which is larger in the self-care domain than in the mobility and cognition subscales. Notably, the self-care subscale is composed of eight items, whereas the other two subscales of only five items and thus are probably less explored.

In this study, we excluded preschool children, which is known to be a vulnerable population given the critical phase of brain development occurring in the first years after birth [86].

Future research is needed to better understand the AF in BT survivors. The identification of the predictors that emerged in the present study should be tested in longitudinal studies as well as in specially tailored rehabilitation programs, for example improving PS and other related neuropsychological domains or providing BT children with compensation tools to reduce the differences between them and their healthy peers.

## 5. Conclusions

This study was the first, to the best of our knowledge, to examine the explanatory effects of different clinical and cognitive factors on the WeeFIM outcomes as a measure of AF in pediatric BT survivors.

Time since diagnosis indicates an important temporal window of several years, in which different dynamics may interplay with one another in influencing AF, possibly related to adjustment processes (clinical and/or psychological). However, the identification of PS as a cognitive predictor suggests a possible target of intervention, together with other important neuropsychological functions related to PS, such as attention, working memory, and/or executive functions.

## Figures and Tables

**Figure 1 cancers-13-04776-f001:**
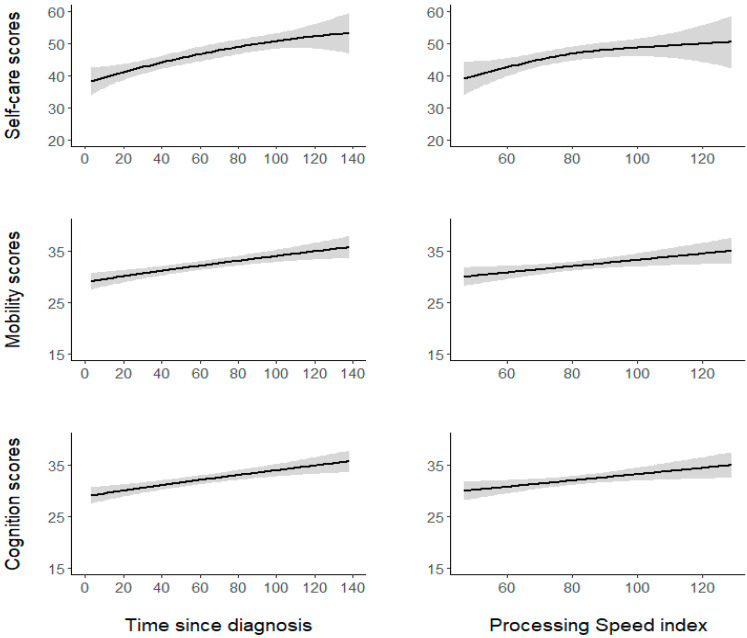
The nonlinear and linear relationships between the significant continuous predictors (time since diagnosis and the PS index) and the WeeFIM self-care, mobility, and cognition subscales. The shaded grey areas represent the SE.

**Figure 2 cancers-13-04776-f002:**
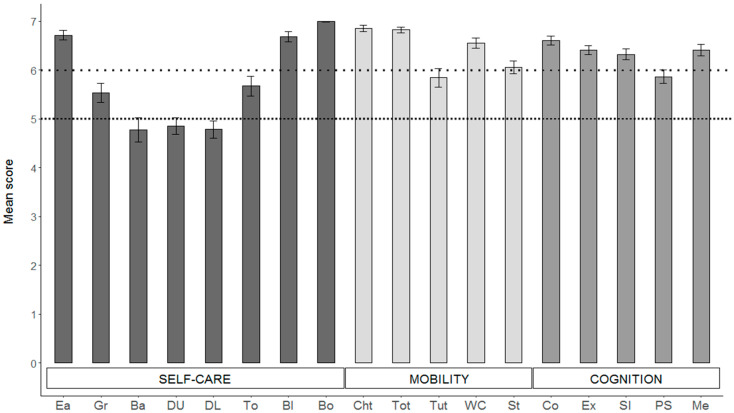
Mean score of the items composing each WeeFIM subscale. Self-care items: Ea = eating; Gr = grooming; Ba = bathing; DU = dressing (upper); DL = dressing (lower); To = toileting; Bl = bladder; Bo = bowel. Mobility items: ChT = bed, chair, wheelchair transfer; ToT = toilet transfer; TuT = tub/shower transfer; WC = walk/wheelchair; St = stairs. Cognition: Co = comprehension; Ex = expression; SI = social interaction; PS = problem solving; Me = memory. Error bars represent 1 SE. Values below the upper dotted line indicate modified independence, while values below the lower dotted line indicate the need for supervision.

**Table 1 cancers-13-04776-t001:** Number (*n*) and percentage (%) or mean (M) and standard error (SE) of clinical variables of the final sample (k = 73) used for statistical analysis.

**Categorical Clinical Variables**	***n* (%)**
Sex	
Male	42 (57.5)
Female	31 (42.5)
Histopathological tumor type	
Astrocytoma	16 (21.9)
Ependymoma	14 (19.2)
Medulloblastoma	28 (38.4)
Others	15 (20.5)
History of hydrocephalus	
Present	13 (17.8)
Absent	60 (82.2)
Tumor location	
Supratentorial	31 (57.5)
Infratentorial	42 (42.5)
Treatment	
Neurosurgery without adjuvant treatments	17 (23.29)
Neurosurgery and chemotherapy	7 (9.59)
Neurosurgery and radiotherapy with or without chemotherapy	49 (67.12)
**Continuous Clinical Variables**	**M (SE)**
Time since diagnosis (months)	59.5 (4.5)
Age at diagnosis (in months)	71.1 (4.6)

**Table 2 cancers-13-04776-t002:** Mean (M) and standard deviation (SD) for the Wechsler Intelligence Scale for Children, 4th Edition indexes. VCI: Verbal Comprehension Index; PRI: Perceptual Reasoning Index; FSIQ: Full Scale Intelligence Quotient; WMI: Working Memory Index; PSI: Processing Speed Index.

Cognitive Variable	M (SD)
VCI	88.41 (18.47)
PRI	87.88 (19.47)
FSIQ	85.88 (18.85)
WMI	88.19 (19.33)
PSI	80.27 (18.02)

**Table 3 cancers-13-04776-t003:** Reports the intercept, standard error (SE), *p*-value, and *R*^2^ adjusted for each model.

WeeFIM Subscales	Intercept	SE	*p*-Value	R^2^ (adj.)
Self-care model	46.34	2.26	<0.0001	0.66
Mobility model	33.78	1.49	<0.0001	0.33
Cognition model	30.87	1.42	<0.0001	0.28

## Data Availability

The data were deposited in the Zenodo database and they are publicly available (doi: 10.5281/zenodo.4733570, Version 001, accessed on 3 May 2021).

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
