# Peer review of "Processing Speed and Time since Diagnosis Predict Adaptive Functioning Measured with WeeFIM in Pediatric Brain Tumor Survivors"

_cancers, 2021, doi:10.3390/cancers13194776_

Round 1

Reviewer 1 Report

I want to thank the authors for the careful revision and the additional analyses!

A careful reading of this version is recommended, eg. numbering in discussion not always correct

Reviewer 2 Report

Oprandi et al have submitted an interesting article regarding the use of the WeeFIM (Functional Independence measure for Children) assessment tool and assessment of other variables to predict adaptive functioning in paediatric brain tumour survivors.

This manuscript appears to have received prior review since the version this reviewer received had many parts of the submitted paper specifically highlighted with a yellow background.

Major concerns

  1. Materials and Methods: Given the significant role played by radiation on neurocognitive outcomes, treatment type classification is highly recommended to be revised to the following: (a) neurosurgery alone; (b) neurosurgery and chemotherapy; (c) neurosurgery and radiation +/- chemotherapy. As this retrospective study has been designed, it is almost impossible to ascertain the significance of radiation as an independent categorical clinical variable. This will necessitate revision of the data analysis accordingly.
  2. Discussion/Conclusions: At no point do the authors address what interventions could be offered once deficits in processing speed are identified. Identifying a problem without offering a potential solution is not as helpful to the readers and those who would adopt the diagnostic approach recommended by the authors.

Specific Comments

  1. Title: It may be helpful to include WeeFIM in the title.
  2. Materials and Methods:  Line 150: How were the 2 patients who did not undergo an operation diagnosed histologically? Did they only have a biopsy? It would be more appropriate to include these two patients in the revised radiation +/- chemotherapy arm to assess the impact of radiation as an independent categorical clinical variable (see above).

Minor Comments

  1. Abstract: Line 27: Replace "were" with "was" twice in the sentence. Line 34: Change "the" to "The".
  2. Introduction: Line 45 and lines 94, 161, 164, 540, 556, 566 elsewhere in the manuscript: Change "competences" to "competencies". Line 50: Add hydrocephalus specifically. Line 51: Replace "miss" with "fail to reach". Line 54: Delete "themselves" since this is redundant with "autonomously". Line 66: Rephrase "of the developmental age". As written, this does not make sense. Line 78: Add a comma after "recurrence [27,28]". Line 106: This is a single sentence paragraph; not recommended.
  3. Materials and Methods: Line 121: What is meant by "biopsy"? Did the authors mean to write "resection" or "surgical" instead? Line 190: Replace "indexes" with "indices". Line 230: Add "tests" after "diagnostic".
  4. Results: Line 374: Delete the letter "f" prior to "controversial". Line 477 (and line 144): Add 'atypical" prior to "teratoid" to conform to WHO accepted terminology. The abbreviation of this tumour is ATRT. Line 479: The quoted survival for glioblastoma is for adults, not children. Line 523: Replace "Problem" with "problem".
  5. Conclusions: Line 591: Add "pediatric" prior to "BT".

Round 2

Reviewer 2 Report

Oprandi et al have comprehensively addressed Reviewer 2's major concerns as well as specific concerns. The addition of a section to the Discussion titled "Hypothesis of Interventions" will be very helpful to the reader.

There remain just a few very minor revisions as follows (using the track changes version of the revised manuscript for page and line number):

  1. Page 4, line 154: Change "undergone" to "undergo".
  2. Page 14, Section 4.5, line 554: Most neurosurgeons would argue convincingly that hydrocephalus is at the very least partly modifiable rather than not modifiable as written.
  3. Line 563: Change to "to restore impaired skills; hence, the implementation...".
  4. Line 574: Revise "Compensation" to "Compensatory".

Author Response

Response to Reviewer 2 Comments

Oprandi et al have comprehensively addressed Reviewer 2's major concerns as well as specific concerns. The addition of a section to the Discussion titled "Hypothesis of Interventions" will be very helpful to the reader.

Point 1: There remain just a few very minor revisions as follows (using the track changes version of the revised manuscript for page and line number):

  1. Page 4, line 154: Change "undergone" to "undergo".
  2. Page 14, Section 4.5, line 554: Most neurosurgeons would argue convincingly that hydrocephalus is at the very least partly modifiable rather than not modifiable as written.
  3. Line 563: Change to "to restore impaired skills; hence, the implementation...".
  4. Line 574: Revise "Compensation" to "Compensatory".

Response 1: We thank the reviewer for careful reading and suggestions that have been applied in the manuscript

This manuscript is a resubmission of an earlier submission. The following is a list of the peer review reports and author responses from that submission.

Round 1

Reviewer 1 Report

This study addresses an important and relevant question about factors that affect adaptive functioning. I acknowledge that research in this population is difficult as many factors are involved and appreciate the efforts to do this kind of studies in this vulnerable group.

Below are some minor and major remarks. If the authors could address these comments it will improve the quality of the study.

  • The introduction is elaborate, could be shortened. The structure could be improved. f.i. authors explain concept AF, underlying cognitive skills for AF, AF in BT, tools to assess AF, predictors AF, tool WeeFIM. Better to combine section on tools and provide arguments why WeeFIM should be used. The authors gave definition of AF and sub-domains, but these subdomains are not same as WeeFIM subdomains. In discussion the authors touch this point, lack of social subdomain. It would be good to see what rationale was to choose for WeeFIM. Additional the authors state that VABS,… are interview based and therefore subject to potential bias. WeeFIM has the same limitation, were there other options. A critical appraisal of AF assessment should be provided in discussion
  • “WeeFIM predictors” better to say predictors of WeeFIM outcomes
  • Major issue is that PS is put on same level in analyses as diagnosis related factors. In introduction the authors state that cognitive skills may act as mediator between clinical variables and AF. PS can also be affected by diagnosis related factors and probably mediate the link. Mediation analyses are more appropriate to investigate this.
  • Diagnosis and treatment related factors are often associated. In the present study the authors have chosen to investigate diagnosis related factors. Given the fact that authors has chosen PS as cognitive function it is somewhat surprising that treatment related factors, i.c. RT is not investigated. There is strong and growing evidence that RT is linked what PS problems.
  • PS is chosen as cognitive skill, because strong correlation with other indices of Wechsler scales in order to avoid multicollinearity and based on important role of PS in BT. These are good arguments, but other important factors such as WM, fluid intelligence, language… are not taken into account. The authors should be more careful with the interpretation of the results, because these other functions were not looked at. The authors stated that several cognitive functions (language,..) were found to be predictive for WeeFIM scores, but these were not assessed. The clinical implication of ‘training PS’ should be stated more carefully. The title is therefore somewhat misleading as it states cognitive predictors, but just one is looked at.
  • Results: it would be interesting to see how the group performed in relation with the Suskauer ea criteria.
  • Discussion is very elaborate and interesting, but should be shortened. E.g. AF and QoL is not assessed in this study. In contrast, the limitation section needs some additional limitations: no treatment related factors, only PS in analysis, impact of QoL, only patients that are testable (so most affected BT are not assessed and for these patient AF will be largely impacted).
  • Time since diagnosis: role of rehab services. Do authors have info about this for their group? If not, would be good to mention it in limitation section.
  • In discussion the authors state that AF could be an independent construct with its own tendencies…? It is unclear what authors mean by this.
  • Table1 age at assessment, at diagnosis and since diagnosis: one of these ages is redundant

Reviewer 2 Report

This is a retrospective cohort study of a mixed population of children (n=78) after being treated for a variety of brain tumours in different regions of the brain.  The authors use regression analysis to seek interactions between patient and clinical factors and adaptive function performance measured by the WEEFIM scale and other interrelated scales of quality of life.

The proposal for the analysis is well justified from the literature, the statistical assessment is well described and appropriately interpreted.  The relatively small cohort of patients wioth a wide variety of patient, clinicla and tumour related variables restricts the value of this study

The main conclusions are that Adaptive Functioning is predominantly determined by processing speed and time since diagnosis, where slowed processing speed and longer time predicts for worse Adaptive functioning.  Additional factors contribute to AF outcomes including hydrocephalus and tumour location.

The observation that impaired processing speed determines poorer adaptive functioning is mechanistically justified by the inevitable impact of slowed processing speed on time-related tests of adaptive functioning.

However the observation that time since diagnosis is associated with worse outcomes is not mechanistically explained.  There is an interaction between age at diagnosis and time since diagnosis in this retrospective cohort which may be contributing to this effect

There is a valuable description of the limitations of the study which focuses on its retrospective nature, small size and mixed tumour and patient related variables.  The study overall is well written and organised. 

Its weakness is the lack of a mechanistic hypothesis to justify the selection of variables selected for study, the exclusion of treatment variables and their impact upon the regression analysis of AF outcome measures being a key weakness.  The treatments involved are associated with known mechanisms of brain injury. 

The impact of posterior fossa syndrome, as a variable identified in patients with posterior fossa tumours, is now well recognised.  42% of this cohort had posterior fossa tumours.  There is no reference to the proportion with posterior fossa syndrome.  Surgical series and multi-centre studies quote 15-28% of patients affected after cerebellar tumour surgery.  Neuro-cognitive outcome studies without identification of this complication are hard to interpret because of the profound impact of PFS on cognitive functioning particularly processing speed.

I would draw the authors' attention to the Toronto experience where PFS and hydrocephalus were the main determinants of neuro-cognitive function in a multiple regression study of a large institution-based cohort of cerebellar medulloblastoma

 (ref: Impact of craniospinal dose, boost volume, and neurologic complications on intellectual outcome in patients with medulloblastoma.

Moxon-Emre I, Bouffet E, Taylor MD, Laperriere N, Scantlebury N, Law N, Spiegler BJ, Malkin D, Janzen L, Mabbott D.J Clin Oncol. 2014 Jun 10;32(17):1760-8. doi: 10.1200/JCO.2013.52.3290. Epub 2014 Feb 10.PMID: 24516024  

This lack of mechanistic justification of variable selection undermines the value of this retrospective cohort study as it means that hypothesis generation is not related to any particular mechanistic variable that could be addressed in future studies to study or reduce the risk of impaired adaptive functioning for the children.    It may be that comparing posterior fossa tumour to supratentorial tumours might be  mechanistically justified for instance  

The conclusion that prolonged intervals after diagnosis offer opportunity for rehabilitation intervention is only justified if there is clear evidence that longer rehabilitation interventions leads to improved restoration of adaptive functioning.  If young age is a factor in prolonged interval then the age at injury may in fact be be the determining factor linked to steps in brain development affected by brain injury at  sensitive stages of brain development.  Consequently  I would suggest that the analysis is reconsidered by selecting mechanistically justified strategy to variable selection and analysis

Reviewer 3 Report

Oprandi and colleagues provide a retrospective review of 73 children with brain tumor, tested for cognitive and AF outcomes on average 5 years after diagnosis.  Although their goal (examining factors associated with decreased AF in brain tumor survivors) is important, they are hampered by a number of methodologic limitations which should be addressed.

Although it can be challenging to gather data on large numbers of brain tumor survivors, the authors are limited by the number of patients and diversity of tumors and presentations.  This has led to a selection of explanatory variables that ignores important variables.  These include:

-Age at diagnosis was not selected as a variable due to its correlation with time from diagnosis to assessment.  This omits an important variable that has previously been shown to be associated with multiple cognitive/functional outcomes.  While the correlation between variables may mean that both cannot be included in a model, it also adds significant doubt to some of the conclusions drawn, since “time to assessment” could be a stand in for “age at diagnosis”.  Therefore, conclusions (line 644) like “Moreover, time since diagnosis indicates an important temporal window for rehabilitation interventions, suggesting that the first phases of the tumor treatments are the ones with the worst outcomes in AF and may necessitate more interventions” might instead mean that earlier age at diagnosis leads to worse outcomes instead.

-treatment variables were also excluded because they correlated with tumor type (particularly astrocytoma).  This means that results and conclusions surrounding tumor type may instead actually reflect differences in treatment.  This is not adequately explained in the manuscript.

-PS was the only cognitive variable considered, due to correlation with other cognitive measures.  Again, the manuscript does not allow for the possibility that processing speed may not be the sole explanatory variable but may represent a wider cognitive disability.  Therefore, targeting PS alone as a modifiable factor may not be appropriate, as in line 647: “On the other hand, the identification of PS as a cognitive predictor suggests a target  of intervention. PS is a potentially modifiable factor with cognitive rehabilitation intervention [125] or with the use of compensatory instruments”

-Finally, from a clinical oncologist perspective, it seems an odd choice to select “time to assessment” and “tumor type” over “age at diagnosis” and “treatment”.  While all of these potential variables are important, I would have preferred to see the latter two.

In addition, the following more minor comments should be addressed:

simple summary:  “

 “Time since diagnosis indicated an important temporal-window for rehabilitation, and processing speed is a potential focus of intervention. By successfully assessing, understanding and managing AF, it will be possible to increase the quality of pediatric BT survivorship.”

I believe the statements above overstate the findings of the study.  “Time since diagnosis” may actually be a stand in for “age at diagnosis”, and even if not, it is not certain that this retrospective study identifies a window for rehabilitation.  PS may be a stand in for cognitive ability in general and improving PS.  Also, there is no direct evidence that managing AF would affect QoL.

Abstract:

 “Conclusions: the main explanatory variables of AF that emerged in our models indicate a potential target and a temporal window for rehabilitation interventions.”

These conclusions seem to overreach the scope of the manuscript.

Materials and Methods:

Hydrocephalus is defined as the presence or absence of hydrocephalus.  It is unclear if this is current hydrocephalus or history of hydrocephalus.  It would be more appropriate for this to be history of hydrocephalus, and this may need to be clarified.

PNET is no longer a WHO diagnosis, and recent publications suggest that many tumors previously classified as PNET actually fit molecular signatures consistent with high grade glioma.  Therefore, this group of tumors may better be classified as the “other tumor” category.

Discussion:

“By successfully assessing, understanding and managing AF, it will be possible to increase the quality of pediatric BT survivorship” 

This is a retrospective study and not able to determine whether managing AF could affect QoL.  This statement should be rephrased to suggest that this is only a possibility.
